# Determining the Characteristics of Farms That Raise Pigs without Antibiotics

**DOI:** 10.3390/ani12101224

**Published:** 2022-05-10

**Authors:** Elise Bernaerdt, Dominiek Maes, Tommy Van Limbergen, Merel Postma, Jeroen Dewulf

**Affiliations:** 1Unit of Porcine Health Management, Department of Internal Medicine, Reproduction, and Population Medicine, Faculty of Veterinary Medicine, Ghent University, Salisburylaan 133, 9820 Merelbeke, Belgium; dominiek.maes@ugent.be; 2ANITOM bv, Pierstraat 122, 2630 Aartselaar, Belgium; tommy.vanlimbergen@anitom.be; 3Veterinary Epidemiology Unit, Department of Internal Medicine, Reproduction, and Population Medicine, Faculty of Veterinary Medicine, Ghent University, Salisburylaan 133, 9820 Merelbeke, Belgium; merel.postma@ugent.be (M.P.); jeroen.dewulf@ugent.be (J.D.)

**Keywords:** pig production, antimicrobial usage, antimicrobial resistance, raised without antibiotics, biosecurity, herd management

## Abstract

**Simple Summary:**

Reduced and responsible antimicrobial use leads to a lower risk of developing antimicrobial resistance. One way to achieve this is to raise animals without antibiotics (RWA). This study described the criteria for a Belgian RWA program for pigs and evaluated whether farms could achieve and maintain this status. The study also identified possible differences between RWA and non-RWA farms. For this purpose, 28 farms were visited three times for the following reasons: (1) data collection, (2) farm-specific coaching, and (3) evaluation. Antimicrobial use, biosecurity, and farm characteristics were determined. The status of the farms, i.e., (non-)RWA, varied over time, and the distribution of RWA vs. non-RWA farms, was 10–18, 13–15, and 12–16 before the intervention, after coaching, and after one year, respectively. There were no significant differences in biosecurity status between RWA and non-RWA farms, but biosecurity improved in all farms throughout the study. RWA farms were smaller (median 200 sows) compared to non-RWA farms (median 350 sows). This study showed that farmers could achieve and maintain the RWA status through farm-specific coaching related to prudent AMU and improved biosecurity.

**Abstract:**

Reduced and responsible antimicrobial use leads to a lower risk of developing antimicrobial resistance. Raised Without Antibiotics (RWA) is a certification label that is recognized in only a few countries, and it is often unclear what the specific criteria and characteristics of RWA farms are. The objectives of this study were to describe the criteria for a Belgian RWA program; to coach farms towards reduced antimicrobial usage (AMU); to assess if it was possible to obtain and maintain the RWA status; and to determine differences between RWA and conventional pig farms. Pig farms (*n* = 28) were visited three times for the following reasons: (1) data collection, (2) farm-specific coaching (2 months later), and (3) evaluation (7 months later). AMU was followed from before the start of the study up to one year after the last visit. AMU, biosecurity (Biocheck.UGent^TM^), and farm characteristics of (non-)RWA farms were compared. RWA was defined as no antibiotics from birth until slaughter. Pigs requiring individual treatment received a special ear tag and were excluded from the program. The status of the farms varied over time, and the distribution of RWA vs. non-RWA was 10–18, 13–15, and 12–16, before intervention, after coaching, and after one year, respectively. For the non-RWA farms, there was a reduction in AMU of 61%, 38%, and 23%, for the suckling piglets, fattening pigs, and sows, respectively, indicating that they were moving toward the RWA status. There were no significant differences in biosecurity status between RWA and non-RWA farms, but biosecurity improved in all farms throughout the study. RWA farms were smaller (median 200 sows) compared to non-RWA farms (median 350 sows). The 4-week system was used more in non-RWA farms, while the 3- and 5-week systems were used most often in RWA farms. This study showed that farmers could achieve and maintain the RWA status through farm-specific coaching related to prudent AMU and improved biosecurity.

## 1. Introduction

Antibiotics are used to tackle infectious diseases caused by bacteria; they can be used for therapeutic purposes and disease control in a herd [1]. However, there is a clear association between antimicrobial usage (AMU) and antimicrobial resistance (AMR) [2,3,4,5]. At a national level, the use of specific antibiotics correlates to the level of resistance toward these antibiotics in commensal *Escherichia coli* isolates in pigs, poultry, and cattle [5]. The problem of AMR has been gaining attention over the years, since AMR can lead to therapeutic failure in both humans and animals, leading to increased morbidity and mortality. Currently, over 700,000 people worldwide die every year due to AMR. If no action is taken, it is estimated that by 2050 this number will increase to 10 million human deaths per year [6,7,8]. The problem of AMR should be addressed with a One Health approach including humans, animals, and the environment [9]. The reduction of antibiotic-resistant bacterial isolates in pig production can be obtained through coaching, better herd management, improved biosecurity, and prudent AMU or the restriction of AMU [4,10,11,12].

Raised Without Antibiotics (RWA) in pig production is a concept that is recognized in only a few countries, such as Denmark, Poland, and the United States. In RWA production, pigs are raised without the use of any antibiotics from birth until slaughter [13,14,15,16,17]. In the United States, RWA is an independent certification that covers all animal source foods including meat, poultry, seafood, fish, dairy, and eggs. It is certified by the National Sanitation Foundation (NSF) [18]. In The Netherlands, there is also an antibiotic-free concept called “Antibioticavrij Leven Garantie” [19]. According to the Danish Integrated Antimicrobial Resistance Monitoring and Research Program (DANMAP), 51 pig farms in Denmark raised pigs without antibiotics in 2018 [17]. Two studies, each investigating two RWA sow farms, further examined RWA production in Danish pig farms [13,15]. However, on a larger scale, it is unclear as to what the characteristics of RWA farms are, as well as which differences exist in comparison to conventional pig farms. Furthermore, the specific inclusion criteria for RWA production are not well specified in the literature, and the implementation of RWA in a larger number of farms with varying management and housing conditions requires further investigation.

Raising pigs without the use of antibiotics is challenging in terms of animal health and welfare. Especially after weaning, piglets are particularly susceptible to various infections, such as *E. coli*, causing post-weaning diarrhea or edema disease, or *Streptococcus suis*, causing bacterial meningitis, septicemia, and polyserositis. Previous studies have shown that most antibiotics in pig herds are used in nursery pigs [13,20,21,22]. The decision as to whether or not to treat these bacterial infections with antibiotics is made after an evaluation performed by the herd veterinarian and the farmer and depends on the severity of the disease. In RWA production, the decision to treat can lead to exclusion from the program. However, RWA should not compromise animal welfare, and the focus should be on the prevention of animal diseases.

The aims of this study were to (1) to describe the criteria for a Belgian RWA program and how it may affect the results of the program, (2) guide pig farmers in the RWA program, assess if it was possible to achieve and maintain the RWA status, and (3) determine the characteristics of the farms that succeeded.

## 2. Materials and Methods

### 2.1. Study Design

Belgian Pork Group is a network of abattoirs and companies active in cutting, deboning, and processing pig meat, and they commissioned the study to develop a product line of pigs raised without antibiotics. The project was presented at their annual meeting with pig producers and farmers who applied voluntarily to participate in the project. To encourage participation in the project, collaborating RWA and non-RWA farms received a monthly incentive of €250 and €125, respectively. In total, 28 pig farmers applied to participate in the study. The geographical distribution of the farms is shown in Figure 1. All farms were located in Flanders, the northern Dutch-speaking part of Belgium.

Researchers of Ghent University performed the study. Due to practical reasons, the study was performed in two consecutive groups. The first group of farms (*n* = 16) was followed between 1 February 2018 and 30 September 2019; the second group (*n* = 12) was followed between 26 August 2019 and 30 June 2021. Both groups were approached and guided using the same methodology.

All farms were visited three times by the same investigator. During the first visit, there was a herd inspection of the different animal categories and the overall farm infrastructure, combined with data collection (see further). Approximately two months later, a second visit was performed. The herd veterinarian was invited to discuss the situation on the farm together with the investigator and the farmer. It was determined as to whether or not the farms could start immediately in the RWA program or if a further reduction of AMU was initially required. Farm-specific recommendations were provided to support the farms in the RWA program or to guide farms towards RWA. During a third visit, approximately seven months after the second farm visit, the situation at the farm was evaluated again and compared to the situation of the first visit. AMU was monitored until one year after the third visit.

### 2.2. Biosecurity, Farm Characteristics, and Performance

The biosecurity status of the farms was determined using the risk-based biosecurity quantification tool Biocheck.UGent^TM^ [23]. This tool allows for making an objective quantification of the biosecurity status of the farm. It is based on a questionnaire of 109 questions in 12 categories and results in a score between 0 and 100%. Zero means a lack of any biosecurity measures, while 100 means perfect biosecurity. For external biosecurity, the following six categories were assessed: (1) purchase of breeding pigs, piglets, and semen; (2) transport of animals, removal of carcasses and manure; (3) feed, water, and equipment supply; (4) visitors and farmworkers; (5) vermin and bird control; and (6) location of the farm. For internal biosecurity, the following six categories were assessed: (1) disease management; (2) farrowing and suckling period; (3) nursery unit; (4) finishing unit; (5) measures between compartments, working lines, and use of equipment; and (6) cleaning and disinfection [24,25,26].

Other farm characteristics—not included in Biocheck.UGent^TM^—were collected during the herd inspection (first farm visit) in a standardized way: name and contact details of the herd veterinarian, herd management characteristics including type of farm, batch management system, sow breed, management and housing of the different animals; i.e., farrowing unit, nursery unit, fattening unit, quarantine unit, and insemination/gestation unit. A specific section was also reserved for all veterinary and non-veterinary treatments, including vaccination protocols, anti-parasitic treatment scheme (product, moment of treatment, and duration), and all feed and water additives.

Performance data for the past year were obtained from the herd management program during the first and third farm visits. The following information was collected for the sows and suckling piglets: number of weaned piglets per sow per year, farrowing index, weaning-to-estrus interval, pregnancy rate, replacement rate, live born piglets, pre-weaning mortality, and weaned piglets per litter. Reproductive data of the sows was available for most of the farms on the first visit, but fewer farms had follow-up information on the third visit. Therefore, the comparison of the performance of the first visit compared to the third visit was not made. For the nursery and fattening pigs, only 11 farms had information on mortality, average daily growth, and feed conversion ratio for the first visit, and only three farms could provide follow-up information for the third visit. Reasons for the loss of follow-up were the use of a different software program or a lack of time to reliably record or extract the data from the program. Therefore, this information was not further analyzed.

### 2.3. Quantification of Antimicrobial Usage

Information on the AMU of the farms was provided by a nationally used database called AB register. AB register is an independent non-profit organization that deals with the registration of AMU for pigs, poultry, and dairy cows. All Belgian veterinarians have to register all antimicrobial use/supply for every farm in this database [27]. Based on the antibiotics used, the monthly BD100 was calculated. The BD100 is a standardized way to quantify the AMU and is nationally used in Belgium [28,29,30]. BD100 is the number of treatment days with antibiotics in 100 days or the percentage of treatment days with antibiotics and it is calculated for the suckling piglets, nursery pigs, fattening pigs, and sows. The numerator of the formula consists of the amount of antibiotics administered (expressed in milligram) and the long-acting factor (LA_bel_), which corrects for products with an active duration longer than 24 h. In the denominator, the Belgian defined daily dose animal was used (DDDA_bel_). These values are defined based on the information in the summary of product characteristics (SPC) for each antibiotic. An overview of the specific values (DDDA_bel_ and LA_bel_) for each product was provided by the Belgian knowledge center on Antimicrobial Consumption and Resistance in Animals (AMCRA) [30]. The total weight of animals at risk for treatment was the average number of animals at the farm multiplied by a standardized weight at treatment (see further). The number of days animals were at risk to receive treatment was also included. Regardless of the number of days at risk, the AMU was always converted to 100 days.
$$BD100=\frac{amount\;of\;antibiotics\;administered\;\left(\mathrm{mg}\right)}{DDDa_{bel}\times \mathrm{kg}\;animal\;‘at\;risk’\times number\;of\;days\;‘at\;risk’}\times LA_{bel}\times 100$$

The average number of animals within each animal category was determined for each farm. For the suckling piglets, this was determined by multiplying the number of sows in a herd by 30, i.e., the average number of weaned piglets per sow per year, divided by 12, i.e., months per year. The standardized weight of the pigs at treatment was defined as 4 kg, 12 kg, 50 kg, and 220 kg for the different animal categories, respectively. The number of days at risk was 30.42, i.e., the average length of a month.

AMU was compared to benchmarking values, which were determined for the different animal categories by the Belgian monitoring system. The benchmarking system includes attention and action values of the BD100 for the different animal categories. These values define three user categories. When the BD100 of an animal category is below the attention value, farms are considered to be low-user farms. They are in the safe zone and there is no need for action plans to reduce AMU. When the BD100 is between the attention and action value, the farms are considered to be attention-users for an animal category. On these farms, extra attention should be paid to AMU, and they should strive for a lower AMU. Finally, when the BD100 exceeds the action value, farms are considered to be high-user farms for an animal category. These farms should immediately take action to reduce AMU. In 2018, these benchmarking values were fixed by AMCRA (Table 1). In the future, these benchmarking values will be further tightened [31].

AMU was determined for three periods per farm (Figure 2). The first period was approximately 14 months before the first farm visit (period A), the second period was the period between the first and the third farm visit (period B), and the third period was the period one year after the last farm visit (period C).

### 2.4. Criteria of the Belgian Raised without Antibiotics Program

Before the start of the study, the inclusion criteria for RWA farms were defined to assure that RWA farms had a low AMU in all animal categories, did not apply any group treatments, and did not use antibiotics prophylactically. In those farms that were identified as RWA, an external company had to perform an annual audit to verify if the farms complied with all RWA criteria.

For the entire farm in an RWA program, prophylactic medication with antibiotics was not allowed. This was defined as the administration of a product to an individual animal or a group of animals without clinical signs to prevent the possible occurrence of an infection. Additionally, group treatments were not allowed. These were defined as any treatment from a therapeutic, metaphylactic, or prophylactic point of view, in which antibiotics were administered to a group of animals orally or parenterally. To comply with the RWA criteria, the BD100 had to be below the attention value of the Belgian benchmarking system for at least three out of four animal categories.

Within a farm that meets the above criteria, only the animals that did not receive any antibiotics from birth until slaughter were labelled as RWA. Pigs originating from sows that received antibiotic treatment could still be raised according to the RWA criteria. If an animal required antibiotic treatment, individual treatments—i.e., parenteral, local, or oral treatments—were allowed. However, correct identification of the treated animals had to be performed carefully through the use of coloured ear tags and forms indicating the antibiotic treatments and the identification and location of the treated animals. When pigs were moved to the next production stage, the treated animals had to be identified and housed separately. For example, suckling piglets that were treated with antibiotics in the farrowing unit had to be housed together in a separate pen in the nursery unit, and nursery pigs that received antibiotic treatment in the nursery unit had to be housed together in a separate pen in the fattening unit. Animals that had received antibiotic treatment were excluded from the RWA program and were slaughtered as conventional pigs. Figure 3 shows a flow chart with the criteria for farms to comply with the Belgian Raised Without Antibiotics program.

### 2.5. Data Analysis

Descriptive statistics were performed for both the continuous and the categorical variables of the farm characteristics. Normality distribution was analyzed graphically via histograms and Q-Q plots. The mean ± SD was calculated for the normally distributed continuous variables; i.e., the weaning age of the piglets, number of pathogens against which animals were vaccinated, weaned piglets per sow per year, farrowing index, weaning-to-estrus interval, pregnancy rate, replacement rate, number of live born piglets, pre-weaning mortality, and the number of weaned piglets per litter. The median, minimum, and maximum values were determined for the not normally distributed continuous variables; i.e., number of sows, AMU (BD100), and biosecurity scores. Percentages were calculated for the categorical variables; i.e., type of farm, origin of the breeding gilts, batch management system, possible castration of the boars, and sow breed.

A parametric independent samples t-test was used to analyze potential differences between groups for the normally distributed data. A non-parametric Mann–Whitney U test was used to analyze potential differences between groups for the not normally distributed data. Levene’s test was used for analyzing the equality of variances. A Fisher’s exact test was used to assess differences between categorical variables. A non-parametric independent-samples Kruskal–Wallis test was used to analyze potential differences within groups for their AMU of the different periods. A multivariate analysis of variance (MANOVA) was performed to analyze potential effects of herd type (RWA vs. non-RWA), study group, and herd size on AMU for the different animal categories. A log transformation of the BD100 was performed to achieve normality distribution. In the model, BD100 was the dependent variable, herd type and study group were the fixed factors, and herd size was the covariate.

A non-parametric Wilcoxon matched-pair signed-rank test was used to analyze potential differences within groups for their biosecurity status on the first farm visit compared to the third farm visit. *p* values below 0.05 were considered to be statistically significant. All statistical analyses were performed using IBM^®^ SPSS^®^ Statistics for Windows Version 24 (IBM Corp., Armonk, NY, USA).

## 3. Results

### 3.1. Compliance with the Belgian Raised without Antibiotics Program

The mean (±SD) duration of periods A and B was 14.1 (±1.0) and 8.9 (±1.0) months, respectively. The duration of period C was for all farms that are the same; namely, 12.0 months (Figure 2).

Taking into account the RWA criteria, farms could be categorized into two groups; i.e., RWA and non-RWA pig producers. Of course, this status could change over time, and non-RWA farms could work towards RWA production. Therefore, the categorization was performed for the three periods; i.e., A, B, and C (Figure 4). Eight farms remained classified as RWA, and 14 farms remained non-RWA during the entire study period. On six farms, the status varied over time (Table 2). These farms will be discussed in more detail in the following paragraph.

Farm 1 was non-RWA in period A but obtained and maintained the RWA status in periods B and C. Initially, the nursery pigs received amoxicillin trihydrate prophylactically (Rhemox premix, 15 mg per kg body weight (BW)) because there were problems with *S. suis* in the nursery unit. Piglets from different sows were housed together in one pen in the nursery unit. The prophylactic use of antibiotics is a reason for exclusion from the RWA program. After farm-specific coaching, the farmers changed the weaning practices, and piglets of the same litter were housed together in a pen in the nursery unit. This allowed the farmer to stop with the prophylactic antibiotic medication of the nursery pigs, and the farm complied with the RWA criteria.

Farm 2 was non-RWA in period A. During this period, suckling piglets were treated with colistin sulphate (Colivet SF 500, 50,000 IU per kg BW) against neonatal diarrhea. Because of this treatment, the AMU of the suckling piglets exceeded the attention value. The attention value was also exceeded for the sows, because of antibiotic treatment in the farrowing unit with procaine benzylpenicillin (Peni-Kel 300,000 IE/mL, 21,000 IU per kg BW). No separate clothing or boots were used for the different stables. Since pathogens can be transmitted indirectly via farm staff, it was advised to use different clothing and boots for the different animal categories. It was especially advised that the suckling piglets in the farrowing unit should be protected, and that the use of boots could help to avoid the transmission of pathogens. For periods B and C, there were no major health issues in the herd. Suckling piglets did not require treatment with colistin sulphate any longer, AMU did not exceed the attention value for this animal category, and the farm could produce according to RWA criteria.

Farm 3 was non-RWA in period A. During this period, the nursery pigs received group treatments with amoxicillin trihydrate (Octacillin 800 mg/g, 16 mg per kg BW) due to problems with *S. suis*. Even though AMU was below the attention value for all animal categories, the group treatment prevented them from producing according to the RWA criteria. In periods B and C, the farmer exchanged group treatments for individual treatments with amoxicillin trihydrate (Duphamox LA, 15 mg per kg BW) only for piglets with arthritis or meningitis. AMU remained below the attention value for all animal categories, making it possible for the farm to produce according to RWA criteria.

Farm 4 was non-RWA in period A since the AMU of two animal categories—i.e., fattening pigs and sows—exceeded the attention value. However, during period A, the farm applied complete depopulation and repopulated with specific-pathogen-free (SPF) sows. The sows were free of porcine reproductive and respiratory syndrome virus (PRRSV), swine influenza virus (SIV), *Mycoplasma hyopneumoniae*, *Actinobacillus pleuropneumoniae*, and *Glaeserella parasuis*. The result of this intervention was not immediately seen in the nursery and fattening pigs since it took some time for the piglets of the SPF sows to reach the nursery and fattening unit. However, after some months, the health status of the farm significantly improved, resulting in fewer infections and less antimicrobial usage. Therefore, the farm could produce according to RWA criteria in periods B and C.

Farm 5 was RWA in periods A and B. During these periods, only AMU of the suckling piglets exceeded the attention value, and no group treatments were given. However, in period C, AMU of two animal categories—i.e., suckling piglets and sows—exceeded the attention value, resulting in the non-RWA status for period C. The main contributor to AMU in suckling piglets in period C was amoxicillin trihydrate (Duphamox LA, 15 mg per kg BW), which was administered against *S. suis* infections.

Farm 6 was RWA in period A. AMU was very low, and there were almost no antibiotic treatments. However, at the end of period B, there were problems in the nursery unit with PRRSV and porcine circovirus type 2 (PCV2). The herd veterinarian decided to treat the nursery pigs with doxycycline hyclate (Doxyral 10% premix, 10 mg per kg BW) and amoxicillin trihydrate (Rhemox premix, 15 mg per kg BW) against secondary bacterial infections, leading to an increase of the BD100, exceeding the attention value. After these problems occurred in the nursery unit, all sows were vaccinated against PCV2, and a few weeks later all sows were intradermally vaccinated against PRRSV. In period C, the results of these vaccinations were not yet visible, and the farm was still unable to produce according to RWA criteria.

### 3.2. Characteristics of RWA and Non-RWA Farms

To compare the characteristics of the RWA farms with the non-RWA farms, the status of the farms for period B and the period between visits 1 and 3 were considered. The comparison of the RWA (*n* = 13) and the non-RWA (*n* = 15) farms was made for farm characteristics, antimicrobial usage, biosecurity, and performance.

#### 3.2.1. Farm Characteristics

The median (min.–max.) number of sows was 200 (85–300) for the RWA farms and 350 (180–1250) for the non-RWA pig farms (*p* < 0.001). The mean (±SD) weaning age of piglets was 24.9 (± 2.6) days for the RWA farms and 23.9 (±2.9) days for the non-RWA pig farms (*p* = 0.360). Two of the RWA farms were SPF for different pathogens; e.g., *Mycoplasma hyopneumoniae*, PRRSV, or swine influenza virus (SIV). Table 3 shows the other farm characteristics of the RWA and non-RWA farms separately.

RWA farms were more often single site farrow-to-finish farms compared to non-RWA farms (*p* = 0.055). RWA farms more often reared their own gilts compared to non-RWA farms (*p* = 0.254). There was a borderline significant association between the use of a specific batch management system and farm status; i.e., (non-)RWA (*p* = 0.058). None of the RWA farms used the 4-week batch management system, while this batch management system was used most often on non-RWA farms. Possible castration of the boars did not seem to influence RWA status (*p* = 0.320). There was a borderline significant association between sow breed and RWA status (*p* = 0.053), and RWA farms seemed to more often use their own crossbred sows.

RWA farms applied fewer vaccinations than non-RWA farms (Table 4). On the RWA farms, 46% (6/13), 85% (11/13), and 92% (12/13) of the farms vaccinated the piglets, gilts, and sows, respectively. The farms not practising gilt or sows vaccination were RWA for the entire study period. On the non-RWA farms, 80% (12/15), 100% (15/15), and 100% (15/15) of the farms vaccinated the piglets, gilts, and sows, respectively.

#### 3.2.2. Antimicrobial Usage

Table 5 shows the median (min.–max.) BD100 of the farms for the different animal categories for periods A, B, and C for the RWA and the non-RWA farms separately. 

First, a comparison of AMU of the different periods was made per animal category, for the RWA and non-RWA farms separately. No statistically significant differences were found. However, for the non-RWA farms, there was a reduction of the BD100 of period B compared to period A for the suckling piglets, the fattening pigs, and the sows, with a decrease of 61%, 38%, and 23%, respectively.

Second, we performed a MANOVA to evaluate the effect of herd type (RWA vs. non-RWA) on the AMU and corrected for the potential effects of study group and herd size. No portion of the analysis study group had a significant effect on AMU. The effect of herd size was in most cases not significant. Only in fattening pigs in period A (*p* = 0.005) and period B (*p* < 0.001) was AMU significantly higher on larger farms, whereas the AMU of sows in period B appeared to be significantly higher on smaller farms (*p* = 0.037) (Table 6). Regarding the herd type, i.e., RWA vs. non-RWA, there was a significant effect of RWA-status on AMU of the following animal categories for the following periods: suckling piglets in period C (*p* = 0.025), nursery pigs in period B (*p* = 0.009) and period C (*p* = 0.015), fattening pigs in period A (*p* < 0.001) and period C (*p* = 0.012), and sows in period B (*p* = 0.004) and period C (*p* = 0.032).

#### 3.2.3. Biosecurity

Table 7 shows the median biosecurity scores (%) of the farms for the different categories of the Biocheck.UGent^TM^ for the first compared to the third farm visit. The scores for the RWA and the non-RWA pig farms are shown separately. The overall external, internal, and total biosecurity scores did not significantly differ between the RWA and the non-RWA pig farms. However, within the two groups, i.e., RWA and non-RWA farms, the overall external, internal, and total biosecurity were significantly better on the third visit compared to the first visit. For both RWA and non-RWA farms, the overall internal biosecurity scores increased by 10%. The biggest improvement was seen in the disease management of the RWA farms, with an increase of a score of 40%. The biosecurity score of the fattening unit and measures between compartments, working lines, and use of equipment improved as well for both RWA and non-RWA farms.

#### 3.2.4. Performance

Table 8 shows the performance parameters of the RWA and the non-RWA farms. This information was collected on the first farm visit. No statistically significant differences were found. Nevertheless, on RWA farms there are less weaned piglets per sow per year compared to non-RWA farms.

## 4. Discussion

This study investigated the implementation of an RWA program applicable to Belgian pig farms. Pig farmers were guided in the program, and the study showed that it was possible to achieve and maintain the RWA status. Furthermore, differences in farm characteristics between RWA and non-RWA herds were elucidated. All three aims of this study were met.

Pig farmers were able to apply voluntarily to this study. Therefore, the described farms are likely not representative of the whole population, as they can be assumed to be more interested and motivated to produce pigs with low AMU. On the other hand, interest and motivation are crucial to produce pigs according to the RWA criteria. Therefore, the presented results can be considered valid for the part of the population that qualifies for inclusion in this type of pig production.

As there is not yet a global agreement on the criteria of RWA production, there are differences in applied criteria between countries. The criteria for Belgian RWA production were drafted at the beginning of 2018. We defined that in RWA production, prophylactic use of antibiotics and group treatments for any reason, including prophylactic, were not allowed. In December of 2018, the new EU regulation on veterinary medicinal products was communicated. One of the goals of this new regulation was to strengthen the EU response to fight AMR, and it was determined that prophylactic use of antibiotics should only be used in exceptional cases for administration to individual animals when the risk for infection is very high or the consequences are likely to be severe. Furthermore, the veterinarian should be able to justify the prescription of antibiotics, especially in the case of metaphylactic and prophylactic use. This new regulation bans the prophylactic use of antibiotics in groups of animals [32]. Our criteria were in line with this legislation, which took effect in January of 2022.

In Denmark, the pig producer decides which piglets are suitable for RWA and provides them with a special ear tag before 4 days of age. If ear-tagged pigs receive a treatment with antibiotics, the special ear tag is then removed, and the pig loses its RWA status [13,15]. In Poland, it is the other way around, and pigs excluded from the RWA program receive an extra ear tag [16]. In the Belgian RWA program, all piglets born in a farm that fulfills the RWA criteria start automatically as RWA, and if a treatment with antibiotics is required, treated pigs get a special ear tag and lose their RWA status. The location of the antibiotic-treated pigs must be known at all times, by using a form indicating identification and location of antibiotic-treated animals. This was included because there is always the risk that a treated animal loses its ear tag. If this happens, it is then made clear by using the form which animals should be excluded from the RWA program.

According to DANMAP, 51 pig farms in Denmark raised RWA pigs in 2018 [17]. In 2018, there were in total 1613 farms with sows in Denmark, meaning that 3% of the Danish sow farms were producing RWA pigs in 2018 [33]. In Belgium, only the farms of this study are known to produce according to RWA criteria. In 2020, there were 1649 sow farms in Belgium. This would mean that currently only 0.7% of the Belgian sow farms produce according to RWA criteria [34].

Antibiotic-free strategies, i.e., the complete restriction of all antibiotics, might be beneficial in reducing AMR, but antibiotic treatments are sometimes very necessary to treat animals that are clinically diseased due to infections with bacterial pathogens. Not treating such animals would have a negative impact on animal performance, farm profitability, and, last but not least, animal welfare [35]. In a study of Baekbo (2017), discontinuation of standard treatment because of the initiation of an RWA program resulted in an increased incidence of umbilical hernia, diarrhea, and arthritis, and the piglets had a slightly lower weight at the end of the nursery unit, illustrating that the introduction of an RWA program is not always feasible or warranted [13]. On the other hand, in the same study, the RWA program did not seem to have a negative impact on the overall productivity of the sow farms, and RWA fattening pigs showed fewer lesions at slaughter; i.e., chronic pneumonia, hernia, and abscesses, compared to non-RWA pigs [13].

In a survey in the United States, 88% of the respondents with RWA experience, and 98% of the respondent with no RWA experience, believed that RWA production slightly or significantly worsens animal health, animal welfare, and food safety, even though antibiotic treatment of sick animals was allowed. Most respondents agreed that more stringent health and welfare auditing is needed when animals are raised without antibiotics, to ensure a good follow-up of animal health [14]. In the RWA program described in the current study, it is possible to treat sick animals at all times, but consequently, such treated pigs lose their RWA status. If the antibiotic treatment does not cause a high increase of the BD100, and the attention value is not exceeded, the farm maintains its RWA status. As such, the animal health and welfare should not be negatively influenced by the program.

In general, most of the antibiotic treatments are administered in nursery pigs [17,36]. The study of Baekbo (2017) on RWA production in two sow farms showed that the nursery period was challenging, and only 69 to 75% of the pigs that were initially selected to participate in the RWA program were still RWA after the nursery period. The most frequently occurring problems were diarrhea, arthritis, and umbilical hernia. Only 38 to 58% of the ear-tagged pigs were finally slaughtered as RWA [13]. In the study of Lynegaard et al. (2021), on RWA production on two farms, similar percentages were found, and 64 and 68% of the pigs reached the end of the nursery period without any antibiotic treatment [15]. This is in line with the findings of the present study, where most antibiotic treatments were administered to the nursery pigs. The exact percentage of pigs that received antibiotic treatment in our study was not known. However, this was expected to be limited, as farms had to comply with the criteria of the RWA program and AMU in general had to be low.

There was an important reduction in AMU for the suckling piglets, fattening pigs, and sows in the non-RWA farms from period A to period B, likely a result of the fact that these farms were guided toward RWA. In the RWA farms, no significant reductions of AMU were observed between period A and B, likely because AMU was already very low, and hence there was not much room for further reduction.

The effect of study group, herd size, and herd type on AMU was evaluated. We found that AMU of the fattening pigs increased with a larger herd size, which is in agreement with several previous studies [37,38]. However, for the sows, we found that larger herds appeared to use less antimicrobials. Yet, the difference in absolute values of AMU in the sows was quite limited. Therefore, it is questionable as to whether the observed effect is also biologically relevant.

Given the importance of the proverb “prevention is better than cure”, the focus should be on the prevention of infectious diseases by improving hygiene and preventing the spread of infection by biosecurity measures [8,39]. In our study, significant improvements in the biosecurity score were established, and the biggest improvements were made for the internal biosecurity. For both RWA and non-RWA farms, the overall internal biosecurity scores increased by 10%. The biggest improvement was seen in the disease management of the RWA farms, with an increased score of 40%. Also, in the study of Baekbo (2017), biosecurity measures were mentioned: enhanced hygiene measurements in the stables and when handling the pigs, the all-in/all-out principle, and changing footwear internally in the farm when different compartments were visited [13].

RWA farms were smaller (median 200 sows) compared to non-RWA farms (median 350 sows). On farms with fewer sows, the groups of nursery and fattening pigs will be smaller. Since direct contact with live animals is the main disease transmission route and a high stocking density can cause a strong rise in the infection pressure [39], it can be expected that the infection pressure decreases on farms with smaller groups of pigs. Subsequently, there is less of a need to treat animals with antibiotics. Therefore, RWA could be mainly feasible on small farms.

An increased weaning age of piglets reduces the risk of developing post-weaning disease, e.g., *E. coli* diarrhea [20], and has been associated with a lower AMU [40,41]. In our study, the weaning age was on average slightly higher on RWA farms (24.9 ± 2.6 days) compared to non-RWA farms (23.9 ± 2.9 days). This is also in line with the Danish approach, where increasing weaning age was part of the RWA strategy [13]. Remarkably, there were no RWA farms using a 4-week batch management system. A possible explanation could be that farms working in a 4-week batch management system wean their piglets usually one week earlier (at max. 21 days) than farms working, for example, in a 3-week batch management system (weaning at 28 days of age). A higher weaning age leads to more robust, resilient, and heavier piglets that have adapted better to solid feed, resulting in a smoother transition at weaning. Furthermore, the performance of piglets weaned at 4 weeks is better [42]. This could lead to less of a need for treatment with antibiotics.

The distribution of farms that did own rearing of breeding gilts and farms that purchased them was 77–23% and 53–47% for RWA and non-RWA farms, respectively. Even though this difference was not significant, it is nonetheless interesting that RWA farms seemed to purchase fewer breeding gilts. Generally, it is assumed that purchasing breeding gilts is a biosecurity risk, as through the introduction of new animals new pathogens also may be introduced, which may lead to more diseases, and subsequently a higher need for antibiotic treatment [39].

There were also no significant differences between RWA and non-RWA farms with regard to the used sow breed. Nevertheless, it is remarkable that none of the RWA farms used the highly prolific Danbred sows, which can give birth to an average of 17.6 liveborn piglets per litter [43]. Along with these large litters, the average birth weight of the piglets is lower compared to sows that produce fewer piglets. Piglets with a lower birth weight show a higher risk of stillbirth and pre-weaning mortality [44]. Moreover, piglets with a lower birth weight show reduced post-weaning performance [44]. These piglets might be more vulnerable to diseases in the nursery or fattening unit, and thus have a higher chance of being treated with antibiotics later in life.

In our study, there was no statistically significant difference between RWA and non-RWA farms for possible castration of the boars, i.e., intact boars, chemical, or surgical castration. However, there were some non-RWA farms using antibiotics prophylactically with surgically castrated boars. These farms could not be included in the RWA program, since RWA farms were not allowed to use antibiotics prophylactically.

We found that RWA farms applied fewer vaccinations compared to non-RWA farms. A smaller proportion of RWA farms vaccinated their piglets, gilts, and sows, and they vaccinated against significantly fewer pathogens. This might seem contradictory since a reduction in AMU can be obtained by increased vaccination [12,45,46]. A possible explanation might be that the health status on RWA farms was higher, resulting in a lower need for vaccination against different pathogens. Furthermore, there were two SPF farms amongst the Belgian RWA farms. On these farms, specific pathogens were absent against which vaccination often occurs in conventional pig farms, e.g., *M. hyopneumoniae*, PRRSV, or SIV. This finding supports an earlier study that indicated that farms vaccinating against more pathogens have a higher AMU from birth until slaughter [40]. 

Finally, there were no statistically significant differences in the performance parameters of RWA and non-RWA farms, suggesting that raising pigs without antibiotics does not necessarily compromise the performance of the pigs. However, we only have information on performance parameters of the sows and pre-weaning piglet mortality, as other information such as mortality rate, average daily growth, and feed conversion ratio of the nursery and fattening pigs was lacking. However, a previous study by Postma et al. (2017) showed that it was possible to reduce AMU without jeopardizing the production parameters [46]. 

## 5. Conclusions

This study showed that it was possible to achieve and maintain the RWA status through farm-specific coaching related to prudent AMU and biosecurity. Characteristics of farms that succeeded were determined, and differences between RWA and non-RWA farms were elucidated. The criteria of the RWA program are clearly described in this study and can be used in other studies. The characteristics of RWA farms can be used to estimate whether farms are suitable to raise their pigs without antibiotics. Further research is needed to reveal the feasibility of RWA on a larger scale.

## Figures and Tables

**Figure 1 animals-12-01224-f001:**
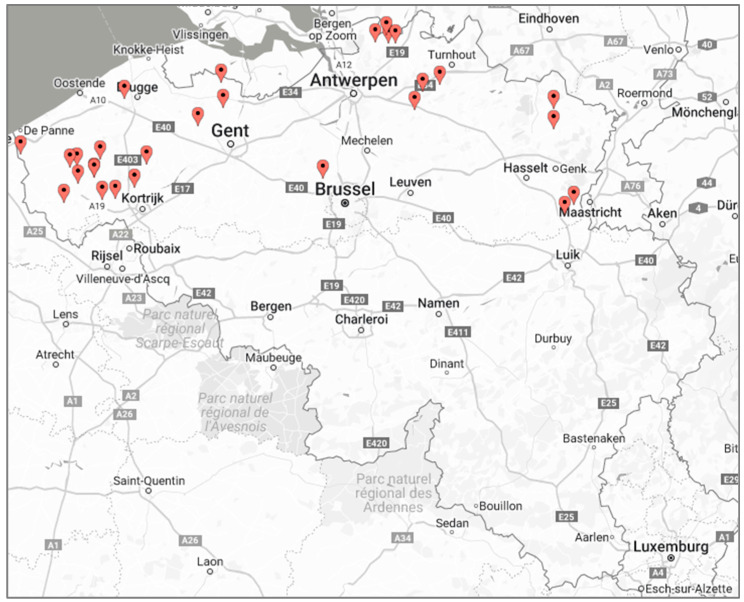
Map of Belgium with the geographical distribution of the farms.

**Figure 2 animals-12-01224-f002:**
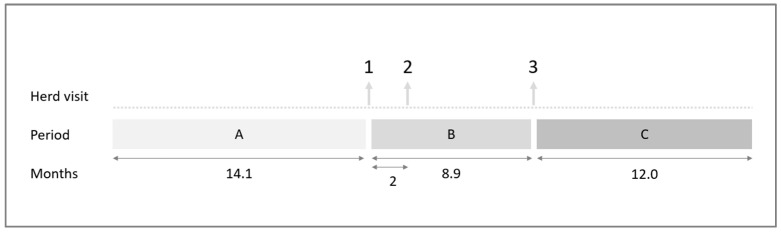
Timeline of the study. Three farm visits were performed, and antimicrobial usage was determined for three different periods (A, B, and C).

**Figure 3 animals-12-01224-f003:**
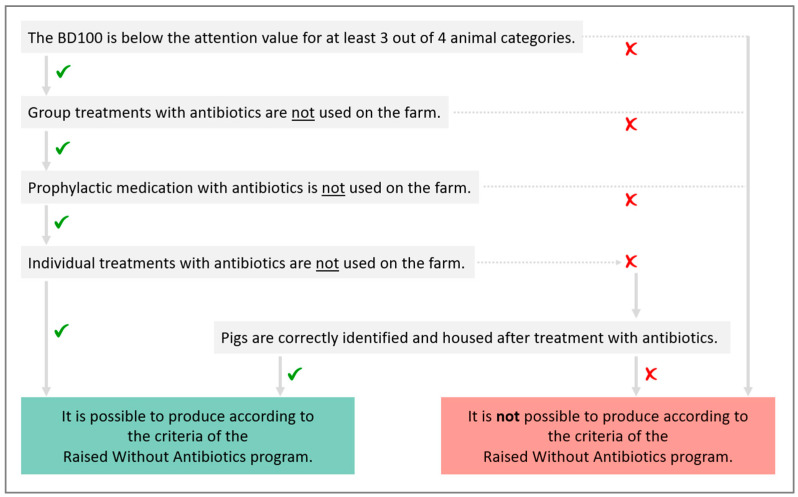
Flow chart to decide if a farm can produce according to the criteria of the Belgian Raised Without Antibiotics program.

**Figure 4 animals-12-01224-f004:**
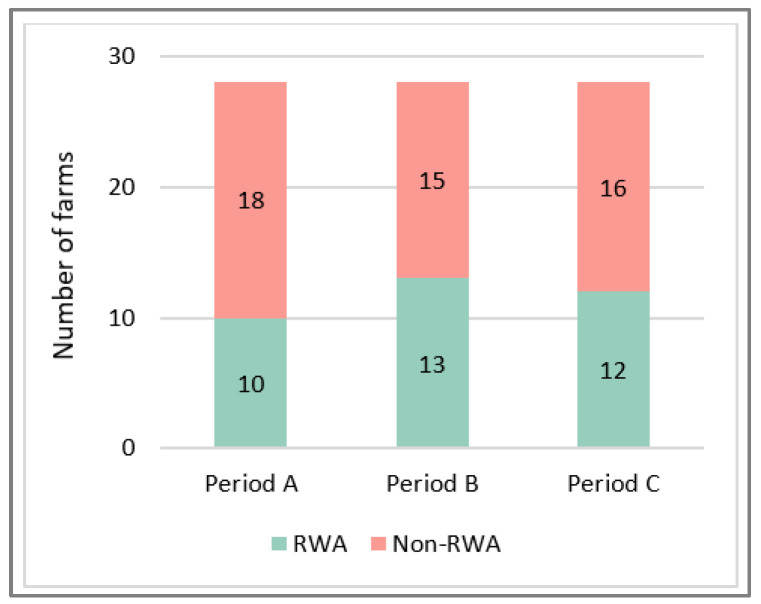
Number of farms producing according to the Raised Without Antibiotics (RWA) program, and number of non-RWA farms in the different periods of the study.

**Table 1 animals-12-01224-t001:** Benchmarking values of the BD100 for the different animal categories (adapted from AMCRA) [30]. To raise pigs according to RWA criteria, the BD100 had to be below the attention value for at least three out of four animal categories.

	Attention Value	Action Value
Suckling piglets	2.00	11.00
Nursery pigs	14.00	51.00
Fattening pigs	2.70	9.00
Sows	0.28	1.65

**Table 2 animals-12-01224-t002:** The status of the farms—i.e., (non-)RWA—could vary over time. For most farms, the status remained the same during the entire study. However, on six farms, the status varied in the different periods of the study (A, B, and C).

	Period A	Period B	Period C
Farm 1	non-RWA	RWA	RWA
Farm 2	non-RWA	RWA	RWA
Farm 3	non-RWA	RWA	RWA
Farm 4	non-RWA	RWA	RWA
Farm 5	RWA	RWA	non-RWA
Farm 6	RWA	non-RWA	non-RWA

**Table 3 animals-12-01224-t003:** Farm characteristics of the Raised Without Antibiotics (RWA) (*n* = 13) and the non-RWA pig farms (*n* = 15).

	RWA (*n* = 13)		Non-RWA (*n* = 15)	
	*n*	%	*n*	%
Type of farm				
*Single site farrow-to-finish*	11	85	7	47
*Multiple sites*	2	15	8	53
Origin of the breeding gilts				
*Own rearing of breeding gilts*	10	77	8	53
*Purchasing of breeding gilts*	3	23	7	47
Batch management system				
*1-week system*	2	15	2	13
*2-week system*	1	8	0	0
*3-week system*	5	38	5	33
*4-week system*	0	0	6	40
*5-week system*	5	38	2	13
Castration of the boars				
*Intact boars*	2	15	6	40
*Castration–chemical*	4	31	5	33
*Castration–surgical*	7	54	4	27
Sow breed				
*Belgian Landrace*	1	8	2	13
*Hypor*	2	15	1	7
*TN70*	3	23	5	33
*Danbred*	0	0	3	20
*Rattlerow Seghers*	0	0	1	7
*Danbred + Rattlerow Seghers*	0	0	1	7
*Danbred + Hypor*	0	0	1	7
*Own crossbred sows*	7	54	1	7

**Table 4 animals-12-01224-t004:** The number of pathogens (mean ± SD) against which piglets, gilts, and sows were vaccinated on the Raised Without Antibiotics (RWA) (*n* = 13) and the non-RWA pig farms (*n* = 15). *p* values are provided for a comparison between RWA and non-RWA farms based on an independent samples *t*-test.

	RWA (*n* = 13)	Non-RWA (*n* = 15)	
	Mean ± SD	Mean ± SD	*p* value
Vaccination piglets	0.7 ± 0.9	1.6 ± 1.1	0.025 *
Vaccination gilts	3.5 ± 1.9	7.7 ± 2.1	<0.001 *
Vaccination sows	4.0 ± 1.8	6.3 ± 1.8	0.002 *

* *p* values below 0.05 were considered statistically significant.

**Table 5 animals-12-01224-t005:** The median (min.–max.) BD100 of the farms for the different animal categories for the Raised Without Antibiotics (RWA) and the non-RWA pig farms. AMU was determined for three periods (A: 14 months before the first farm visit; B: between first and third farm visit; C: one year after third farm visit). The distribution of RWA vs. non-RWA farms was 10–18, 13–15, and 12–16, for periods A, B, and C, respectively. To raise pigs according to RWA criteria, the BD100 had to be below the attention value for at least three out of four animal categories.

	Attention Value	RWA						Non-RWA					
		Period A		Period B		Period C		Period A		Period B		Period C	
Suckling piglets	2.00	0.15	(0.00–26.10)	0.02	(0.00–10.10)	0.00	(0.00–1.28)	4.04	(0.00–78.40)	1.56	(0.00–34.48)	3.93	(0.00–24.73)
Nursery pigs	14.00	0.82	(0.35–15.56)	0.82	(0.00–29.18)	1.15	(0.05–10.27)	12.04	(0.02–75.90)	14.55	(1.20–97.78)	13.40	(0.00–88.78)
Fattening pigs	2.70	0.07	(0.00–0.95)	0.07	(0.00–1.86)	0.10	(0.00–0.56)	2.50	(0.11–9.66)	1.54	(0.00–5.48)	2.32	(0.00–11.20)
Sows	0.28	0.11	(0.00–0.83)	0.18	(0.00–2.23)	0.16	(0.00–1.02)	0.61	(0.00–5.98)	0.47	(0.02–12.71)	0.59	(0.00–5.34)

**Table 6 animals-12-01224-t006:** The median (min.–max.) BD100 of the animal categories where a significant effect of herd size on AMU was found. Farms were categorized into two groups; namely, farms with a herd size smaller than the median for the corresponding period and farms with a herd size equal to or larger than the median.

	Herd Size < Median	Herd Size ≥ Median
Fattening pigs (period A)	0.77 (0.00–9.66)	1.97 (0.01–8.92)
Fattening pigs (period B)	0.09 (0.00–4.24)	1.59 (0.00–5.48)
Sows (period B)	0.39 (0.00–12.71)	0.18 (0.00–4.21)

**Table 7 animals-12-01224-t007:** The median (min.–max.) biosecurity scores (%) of the farms for the different categories of the Biocheck.UGent^TM^ survey for the Raised Without Antibiotics (RWA) (*n* = 13) and the non-RWA pig farms (*n* = 15). The survey was filled in during the first and third farm visit, jointly by the researcher and the farmer. *p* values are provided for a comparison between RWA and non-RWA farms based on a non-parametric Wilcoxon matched-pair signed-rank test.

	**RWA (*n* = 13)**					**Non-RWA (*n* = 15)**				
	**Visit 1**		**Visit 3**		***p* value**	**Visit 1**		**Visit 3**		***p* value**
**External biosecurity**	**66**	**(52–89)**	**71**	**(59–89)**	**0.005 ***	**70**	**(54–84)**	**72**	**(57–87)**	**0.002 ***
*Purchase of breeding gilts, piglets, and semen*	88	(56–100)	88	(60–100)	0.655	88	(76–100)	88	(78–100)	0.157
*Transport of animals, removal of carcasses and manure*	78	(39–87)	83	(70–90)	0.005 *	78	(39–87)	83	(43–95)	0.012 *
*Feed, water, and equipment supply*	33	(17–90)	37	(17–90)	0.180	37	(27–67)	40	(27–67)	0.180
*Visitors and farmworkers*	65	(35–100)	65	(47–100)	0.066	76	(65–100)	76	(65–100)	0.034 *
*Vermin and bird control*	60	(30–100)	60	(30–100)	1.000	70	(30–100)	70	(30–100)	0.317
*Location of the farm*	70	(30–100)	70	(30–100)	1.000	40	(20–100)	40	(20–100)	1.000
**Internal biosecurity**	**48**	**(24–87)**	**58**	**(30–87)**	**0.005 ***	**53**	**(32–76)**	**63**	**(32–85)**	**0.018 ***
*Disease management*	40	(40–100)	80	(40–100)	0.025 *	40	(40–100)	40	(40–100)	0.109
*Farrowing and suckling period*	64	(21–100)	71	(21–100)	0.068	57	(29–100)	71	(29–100)	0.109
*Nursery unit*	71	(36–100)	71	(43–100)	0.109	57	(14–86)	64	(14–86)	0.180
*Fattening unit*	64	(21–100)	79	(36–100)	0.042 *	75	(36–100)	86	(36–100)	0.180
*Measures between compartments, working lines,* *and use of equipment*	32	(7–100)	50	(7–100)	0.018 *	39	(18–86)	50	(18–86)	0.043 *
*Cleaning and disinfection*	50	(0–98)	50	(0–98)	0.180	65	(20–95)	65	(20–95)	0.059
**Total biosecurity**	**56**	**(47–88)**	**64**	**(51–88)**	**0.005 ***	**62**	**(43–78)**	**65**	**(45–86)**	**0.002 ***

* *p* values below 0.05 were considered statistically significant.

**Table 8 animals-12-01224-t008:** The mean ± SD values of the performance parameters on the Raised Without Antibiotics (RWA) (*n* = 13) and the non-RWA pig farms (*n* = 15). This information was collected on the first farm visit. *p* values are provided for a comparison between RWA and non-RWA farms based on a parametric independent samples *t*-test.

	RWA (*n* = 13)		Non-RWA (*n* = 15)	
	*n*	Mean ± SD	*n*	Mean ± SD
Weaned piglets per sow per year	12	27.40 ± 3.60	15	28.97 ± 4.42
Farrowing index	12	2.34 ± 0.14	15	2.32 ± 0.12
Weaning-to-estrus interval (days)	8	6.21 ± 1.53	11	5.67 ± 0.73
Pregnancy rate (%)	9	94.24 ± 4.25	10	90.16 ± 5.38
Replacement rate (%)	10	44.10 ± 12.70	11	46.53 ± 10.79
Live born piglets	13	13.48 ± 1.18	15	14.41 ± 1.98
Pre-weaning mortality (%)	12	13.73 ± 5.28	15	13.29 ± 4.82
Weaned piglets per litter	12	11.62 ± 1.00	15	12.45 ± 1.51

## Data Availability

The datasets used and/or analyzed during the current study are available from the corresponding author upon reasonable request.

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
