# Peer review of "Determining the Characteristics of Farms That Raise Pigs without Antibiotics"

_animals, 2022, doi:10.3390/ani12101224_

Round 1
Reviewer 1 Report
The title as well as keywords accurately reflects the major findings of the work.
The abstract adequately summarize methodology, results, and significance of the study. However, Authors should indicate the age of animals enrolled in the survey as well as body weight.
Also, Authors wrote “Pig herds (n = 28) were visited three times…”, please indicate what the visit is about. Moreover, the statistical analysis applied on data should be briefly indicated.
The introduction section is well written and it falls within the topic of the study. However, Authors should add the concept of One Health emphasizing the importance to reduce the use of antibiotics in livestock as a useful tool to increase both animal welfare and public health.
The section of Materials and Methods is clear for the reader and it meticulously describes the methods applied in the study. However, Authors should check this section and correct many punctuation errors. Moreover some missing information should be added:
Regarding statistical analysis, Authors wrote “A parametric independent samples t-test was used to analyze potential differences between groups for the normally distributed data. A non-parametric Mann-Whitney U test was used to analyse…”. Thus, I understand that Authors checked the Gaussian distribution of data by the application of normality test. Please, indicate the normality test applied in this section before the sentence “A parametric independent samples t-test was used to analyze potential differences between groups for the normally distributed data. A non-parametric Mann-Whitney U test was used to analyse…”
Results section as well as Discussion section is clear and well written. The findings obtained in the study were well discussed and justified with appropriate references.
The conclusion section should be improved. Authors should better summarize the results and the significance of the study.
The tables as well as the figures are generally good and well represent the results of the study.
Authors should check and standardize the references in the list according to journal guidelines.
Reviewer 2 Report
Dear Authors,
I was very pleased when the Editor asked me to review your piece of work. I found it stimulating as it adds new knowledge to a topic of great interest and topicality.
In general, you provide sufficient background to the comprehension of the objectives of the study which are clearly stated. However, the first item, namely “to define the inclusion criteria for an RWA program applicable to the Belgian pig sector”, seems to me already met and not under discussion in the present study. Furthermore, the objectives are not addressed individually in the discussion chapter.
The farms are well described, in detail, and so are the methods. You provide plenty of information on how the indicators were calculated, but some statistical tests are not appropriate for the purpose.
Not all the results you commented on in the discussion are reported in full in the results chapter. Table captions should be improved to make them stand alone. Some of your conclusions are not consistent with the results. The main issue is the analysis you made on ABU comparing RWA and non-RWA farms since ABU is the categorization variable used to define RWA.
I suggest adopting a “STrengthening the Reporting of OBservational studies in Epidemiology” (STROBE) approach/checklist to uniform and strengthen the scientific soundness of your work.
Below, I go through the text with detailed comments.
Title page
Line 2. Please, consider a more descriptive title. The one you chose catches the attention, but do not add any information about your work. Since the title is, sadly, the only thing many researchers read of a manuscript, a more descriptive one would help understand the contents.
Line 10-11. This statement is controversial. I would cautiously say that reduced and responsible antibiotic use leads to a lower risk of developing antibiotic resistance.
Line 23. Please, see the previous comment.
Line 24. The RWA concept is known not only in the few countries you mentioned but also elsewhere. Maybe, you can replace the term “known” with “established” or “recognized”.
Introduction
Line 50. ABR, standing for Antibiotic Resistance, is a rather niche acronym. I suggest using the more popular AMR (antimicrobial resistance) to increase the chances of your work getting cited. Also, in another article published on the same topic and in the same special issue you are applying, the authors used AMR instead of ABR.
Line 55. This statement is a bit unclear. Where do 700,000 people die every year due to AMR? Please, provide a reference or add details to your statement.
Line 61. Please, see the comment on line 24. I would not say that the RWA concept is known only in Belgium, Denmark, Poland and USA. You can replace “known” with “established” or “recognized”.
Line62. Although being an industrial production, pigs are still raised and not manufactured. Consider changing the term “produced”.
Line 63. Consider replacing “animal foods” with “animal source foods”.
Materials and methods
Line 101. In pig farms, is the incidence of bacterial diseases the same between February and September and between August and June? If not, explain why readers shouldn't worry about the discrepancies between the two groups.
Line 105. Please, be more precise about the meaning of “low ABU” in terms of the number of treatments.
Line 108. Consider rephrasing this and other passive sentences in the active form to improve clarity.
Line 109. Have you assessed the reproducibility of the assessment by comparing the result of different investigators in the same herd? Or did the auditing company have a quality standard assessment? If so, please provide further details. Following the same herd for a few months could introduce bias based on the relationship between the inspector and the farmer. This can be a serious flaw in the reproducibility of your study and jeopardise the possibility of further comparisons.
Line 141. Please, consider depositing the questionnaire in an open access repository like Zenodo or arXiv, to assign it a DOI, make it citable, and improve reproducibility.
Line 148. Please, briefly provide the reason for losing the herds at follow-up. Are the herds lost different from the other herds? Could this loss introduce bias?
Line 155. Is there any official reference for the AB-register? If so, please provide it.
Line 188. Please, explain clearly what the Attention and Action values mean.
Line 193. Why is the precedent period longer than the follow-up?
Line 208. The use of the chi-squared test is inappropriate when the expected values are lower than 5 in at least one cell. You should use another test of independence like Fisher’s exact or maybe an alternative statistical approach.
Line 214. The definition of statistical significance risks hampering the results, especially considering the small sample size you had. Please have a look at Amrhein et al. 2019. Nature, 567: 305-307.
Results
Line 243. In Figure 2, you chose to use the “yes” answers as the path that leads to RWA. I agree with your choice, although it requires using negative questions. Since they reduce readability and might create misunderstandings, please state that negative questioning was adopted in the caption.
Line 263. In table 2, you reported herd classification. The initial Herd 4 was not the same Herd 4 which became RWA since all sows were replaced during the study. On the other hand, the structures and the personnel remained the same across the study period. The term “herd” refers to the group of animals, while “farm” includes structures, personnel etc. Please consider those definitions when addressing the classification in RWA and non-RWA and accordingly revise “herd” and “farm” usage in the whole manuscript.
Line 329. Which tests did you use to analyse herds’ characteristics? I tried chi-squared and Fisher’s exact tests using the frequencies in Table 3 and the p-values did not match with yours. In particular, I found more “statistically significant” differences. I used STATA 17 (STATA Corp., College Station, TX, USA).
Line 337. Statistically speaking, the difference you found was not significant. However, that doesn’t imply that there was no difference. I encourage you to examine the data from a veterinary point of view.
Line 339. In Material and methods, you stated that you used the chi-squared test. However, in this case, it was inappropriate since in many cells the expected values were below five. I also suggest the use of (exact) logistic regression to estimate odds ratios with confidence intervals as they give interesting insights.
Line 352. The statement "The P value is statistically significant (P <0.05)" appears to be a fact, but it is not. Please indicate that 0.05 is a threshold of your choice as it is not prescribed by any statistical law.
Line 361. I’m glad to see this paragraph where you analyse results based on your professional competence regardless of the statistical significance. Please, maintain this attitude when examining all the results.
Line 365. The use of "borderline significant" is encouraged by some statisticians (please take a look at http://blog.pmean.com/borderline-significance/). However, here you compare the ABU between RWA and non-RWA and conclude that it differs. I think this result is evident. The flaw is that you compare categories based on the categorization variable you used to classify them yourself. What's worse, based on the statistics, the RWA categorization would not be able to differentiate between high and low ABU in suckling piglets. Please consider your conclusion carefully.
Line 374. Since the same investigator performed visits 1, 2 and 3, the results could be biased and hardly comparable among different herds. I don't doubt the professionalism of the investigators, but the relationship between them and the breeders could bias the assessment. However, it might be interesting to compare the differences in RWA and non-RWA scores. Has one group grown more than the other?
Line 379. See the comment on line 352.
Line 379. In table 6, P values are reported for all Visit 1 – Visit 3 comparisons. It could be appropriate to apply a correction for multiplicity. When testing the same sample for more than one parameter, you have a higher chance of finding significant differences, and P values should be adjusted accordingly. As a reference, please see Payne J.L., 1974. Polity, 1: 130-138 (digitally available at https://www.jstor.org/stable/3234273). Here, I would suggest using a Bonferroni correction or Q value (Storey J.D., 2003. The Annals of Statistics, 31(6): 2013-2035).
Line 383. Regardless of statistical significance and based on your professional competence, are there any differences?
Line 386. Please, see the comment on line 352.
Discussion
Line 400. Please, see the comment on line 50.
Line 425. Please, see the comment on line 50.
Line 470. Please, avoid speculations about unobserved events regardless of the likelihood that they will happen in the future.
Line 478. The 10% increase was only presented as raw values in Table 6. Please, always report explicitly the results you discuss in the further chapter.
Line 481. See the previous comment. If you deem those results interesting enough to be discussed, please consider reporting them in full in the results chapter.
Lines 489-492. This claim is not supported by the findings. First, you are linking batch systems at weaning age, although they are different parameters (a five-week BMS can work either with a 21- or 28-day weaning). Secondly, the test you used was not appropriate when you have cells with zeros. On the other hand, it is noteworthy that no RWA herd has chosen a 4 week BMS. In your opinion, what is the reason? In other words, your results do not support the existence of an association, but if there was, what could it be determined by?
Line 496. Please, see the comment on line 361.
Line 509. Did you evaluate the number of piglets per litter in comparison to the weaning age? Could it support your statement?
Line 530. Lacking information usually requires a sensitivity analysis. What would happen if the lacking information pointed in the opposite direction? How low the fertility rate should be (or high the replacement rate) to make RWA farms perform significantly worse than non-RWA farms? Are those hypothesised values credible?
Conclusions
Line 538. In my opinion, you provided quite an exhaustive explanation of the feasibility of RWA. Why do you believe it is not enough to be extended to a larger scale?
In conclusion, I will suggest that the editor reconsider your manuscript after major revisions. The work is really interesting, however, it needs to be revised in some methodological parts.
Reviewer 3 Report
Dear authors,
Your article is very interesting. It deals with a current topic which is the use of antibiotics in animal production. The manuscript is clearly written, the results are well presented and the discussion is well conducted. I think your paper can be published after some minor corrections.
Please find my comments below:
-Add a map of the distribution of the farms involved in your study.
-Explain further why you did not take into consideration the study period (2018-2019 and 2019-2021) on the results obtained.
-The herd size of farms not using antibiotics is small compared to farms using antibiotics. Are the results obtained in your study more affected by husbandry practices associated with small herd sizes or by the use of antibiotics?
-Concerning the statistical analyses, is it possible to introduce the effects of the period of the study and the size of the farm?
Round 2
Reviewer 2 Report
I am pleased to have revised this manuscript which I believe is of high scientific interest for the subject matter. The authors have answered all my doubts comprehensively and made the presentation of the work clear. All problems of a statistical nature have been adequately addressed, therefore, I suggest that the editor accept this manuscript in its current form, without the need for further modifications.
Author Response
The authors would like to thank the reviewer for the positive evaluation of the revised manuscript.